

# Organic carbon content in arable soil – aeration matters

Tino Colombi[1,2,*], Florian Walder[2,*], Lucie Büchi[3,4], Marlies Sommer[2], Kexing Liu[2,5], Johan Six[6], Marcel G. A. van der Heijden[2,7,8], Raphaël Charles[3,9], Thomas Keller[1,2]

[1]Department of Soil and Environment, Swedish University of Agricultural Sciences, Uppsala, 11824, Sweden
[2]Department of Agroecology and Environment, Agroscope, Zurich, 8046, Switzerland
[3]Department of Plant Production Systems, Agroscope, Nyon, 1260, Switzerland
[4]Natural Resources Institute, University of Greenwich, Chatham Maritime, ME4 4TB, United Kingdom
[5]College of Natural Resources and Environment, South China Agricultural University, Guangzhou, 510640, China
[6]Department of Environmental Systems Science, ETH Zurich, Zurich, 8092, Switzerland
[7]Department of Evolutionary Biology and Environmental Studies, University of Zurich, 8092, Zurich, Switzerland
[8]Plant-Microbe Interactions, Faculty of Science, Institute of Environmental Biology, Utrecht University, 3584, Utrecht, Netherlands
[9]Department of Extension, Training and Communication, Research Institute of Organic Agriculture FiBL, 1001, Lausanne, Switzerland

*These authors contributed equally

Correspondence to: Tino Colombi (tino.colombi@slu.se)

**Abstract.** Arable soils may act as a sink in the global carbon cycle but the prediction of their potential for carbon sequestration remains challenging. The role of soil structure and related physical properties for carbon sequestration is only little explored, especially at the farm level. We hypothesized that improved soil aeration, which is strongly controlled by soil structure, leads
to higher soil organic carbon content. Soil gas transport properties, water holding capacity, microbial biomass and soil organic carbon content, were quantified in the topsoil and subsoil in 30 fields of individual farms. The fields were managed either conventionally, organically or according to no-till practice. Tillage significantly increased gas transport capability and water holding capacity of the topsoil. In the same soil layer, organic farming resulted in higher soil organic carbon content and microbial biomass. Both in the topsoil and the subsoil higher gas transport capability and water holding capacity led to
increased soil organic carbon content ($0.53 < R^2 < 0.71$). In turn, increased organic carbon content improved soil physical conditions. We therefore propose a positive feedback cycle between soil structure and related physical properties like gas transport properties and water holding capacity, root growth, organic matter input and soil organic carbon content. Additionally, higher gas transport capability also increased microbial biomass in the topsoil ($0.50 < R^2 < 0.57$), underscoring the crucial role of soil aeration for soil carbon cycling. Our findings demonstrate the importance of soil aeration for carbon
storage in soil and thus highlight the need to consider aeration in the evaluation of carbon sequestration strategies in cropping systems.





## 1 Introduction

Arable soils play a crucial role in the global carbon cycle because they can act both as a terrestrial carbon sink and source (Lal, 2004a; Smith et al., 2008; Zomer et al., 2017). The conversion of natural vegetation into arable land leads in most cases to a loss of soil organic carbon, and agricultural intensification may aggravate these effects (Balesdent et al., 2000; Celik, 2005;

Głab et al., 2009; Guo and Gifford, 2002; Lal et al., 2011). A decline in soil organic carbon content may ultimately result in decreased water holding capacity, soil structural instability, disrupted water, nitrogen and phosphorus cycles, and eventually decreased crop productivity. Furthermore, the loss of organic carbon from soil contributes significantly to increased atmospheric carbon dioxide concentrations and thus global warming (Lal, 2004b, 2004a). The adoption of different soil management approaches such as diverse crop rotations, reduced tillage intensity, cover crops and the use of organic

amendments bears the potential to alleviate the adverse impacts of agriculture on global soil organic carbon stocks (Diacono and Montemurro, 2010; Freibauer et al., 2004; Kuhn et al., 2016; Lal, 2004a; Merante et al., 2017; Smith et al., 2008). However, discordant results on the effects of specific soil management options on carbon contents have been reported for different cropping systems (Chan et al., 2003; Gattinger et al., 2012; Govaerts et al., 2009; Leifeld et al., 2009; Powlson et al., 2014), which merits further investigation to better understand factors and processes that control soil organic carbon contents in

cropping systems.

Plants convert carbon dioxide into organic carbon through photosynthesis. A large part of soil organic carbon is derived of roots in the form of root biomass and root exudates (Balesdent and Balabane, 1996; Kätterer et al., 2011; Kong and Six, 2010; Rasse et al., 2005). Root growth is greatly affected by soil structure and related physical properties such as soil aeration, water holding capacity and soil penetration resistance. High penetration resistance, low water holding capacity as well as poor soil

aeration and the resulting hypoxic conditions result in decreased root growth (Bengough et al., 2011; Jin et al., 2013; Rich and Watt, 2013; Valentine et al., 2012). Soil aeration is controlled by gas diffusivity through diffusive transport and air permeability through advective transport. Both gas diffusivity and air permeability are key properties that constitute the soil physical constraints on soil aeration (Horn and Smucker, 2005). Gas transport properties are often estimated from proxy values such as total and air-filled porosity. However, because these proxies do not account for pore connectivity, which is crucial for air

circulation and thus oxygen concentration in soils, they do not accurately describe gas transport in soil. Therefore, relative gas diffusion coefficients and air permeability need to be quantified to assess gas transport capabilities of soil. Root growth slows down with decreasing soil oxygen concentration, and it has been shown that roots preferentially grow towards well aerated soil compartments (Colombi et al., 2017; Dresbøll et al., 2013; Porterfield and Musgrave, 1998; Thomson et al., 1992; Watkin et al., 1998). Apart from promoting root growth and hence soil organic carbon input into soil, increased gas transport capability

of soil may also lead to higher microbial biomass and activity, and thus faster decomposition of soil organic matter (Balesdent et al., 2000; Keiluweit et al., 2016, 2017; Young et al., 1998).

Soil structure and associated soil physical properties are greatly affected by soil management. Tillage has been shown to improve penetrability and aeration of the topsoil (Colombi et al., 2018; Dal Ferro et al., 2014; Martínez et al., 2016b, 2016a;



Schjønning and Rasmussen, 2000). However, long-term tillage can also lead to decreased gas transport capability and high penetration resistance both in the topsoil (Kahlon et al., 2013) and the subsoil (Martínez et al., 2016b, 2016a). Diverse crop rotations that include ley and deep rooting species such as rapeseed and oil-seed radish increase soil porosity (Chen et al., 2014; Lesturgez et al., 2004; Stewart et al., 2014; Young et al., 1998), which results in better aeration, increased water holding

capacity and decreased penetration resistance. Permanent soil cover and organic amendments have been shown to increase water infiltration and soil water retention and to improve soil aeration and penetrability (Albizua et al., 2015; Bronick and Lal, 2005; Kahlon et al., 2013). Several studies showed that high soil organic carbon content coincides with good soil aeration, high water holding capacity and low penetration resistance (Albizua et al., 2015; Celik, 2005; Diacono and Montemurro, 2010; Kahlon et al., 2013; Martínez et al., 2016b, 2016a; Rasool et al., 2007; Reynolds et al., 2008; da Silva et al., 2014). These

studies were however limited to one or a few field sites and thus covered only a small diversity of soil textures, which makes it difficult to link soil physical properties to soil organic carbon content. Moreover, the results were obtained from field plot studies where agricultural management may differ substantially from the conditions on commercial farms.

Soil physical properties in general and soil aeration in particular were proposed to play a key role in the regulation of carbon cycling in arable soil (Qi et al., 1994). This is strongly supported by the close interrelation between soil aeration and root

development (Dresbøll et al., 2013; Porterfield and Musgrave, 1998; Thomson et al., 1992; Watkin et al., 1998; Young et al., 1998), and the influence of soil aeration on microbial activity and therefore decomposition rates of organic matter (Balesdent et al., 2000; Keiluweit et al., 2016, 2017; Young et al., 1998). Quantitative information about the relationships between soil gas transport capability and soil organic carbon contents in cropping systems is however limited, and soil aeration is typically not included in soil quality assessments (Bünemann et al., 2018). To gain a better understanding about the potential of soil

management approaches to contribute to carbon sequestration in arable soils, the role of soil aeration has to be investigated. To understand the interrelations between soil management, aeration and organic carbon content at relevant scales and under realistic management conditions, on-farm studies carried out at multiple sites are needed. In the current study we tested the hypothesis that better soil aeration results in higher organic carbon contents. The study was conducted on 30 fields of individual farms in the eastern part of Switzerland. The fields were managed according to three different management systems: i)

conventional farming without tillage, ii) conventional farming with tillage and iii) organic farming with tillage. Gas diffusivity, air permeability and soil organic carbon content were quantified in the topsoil and the subsoil. Additionally, soil penetration resistance, total and air-filled porosity, soil textural composition, water holding capacity, and microbial biomass and respiration were assessed.

## 2 Material and Methods

### 2.1 Study design

The study was performed on 30 separate fields of at least 1 ha in size each, which belonged to 30 individual farms in the eastern part of Switzerland. The fields were managed following three different management systems: conventional (integrated)



farming with no-till practice since at least five years, conventional (integrated) farming with tillage since five or more years, and organic farming with tillage since at least five years. In the remainder of this paper, we refer to the three different management systems as "no-till", "conventional" and "organic", respectively. The system of integrated farming in Switzerland includes a number of practices aiming to enhance the sustainability of cropping systems, which are summarized as the "Proof of Ecological Performance". Farmers need to comply with the "Proof of Ecological Performance" in order to receive full state subsidies. It includes an even nutrient balance, a diverse crop rotation including crops from different botanical families like small grain cereals, maize, rapeseed, and grain and forage legumes, as well as the avoidance of bare fallow soil and targeted use of plant protection products (Swiss Federal Council, 2014). Ten farms per management system were selected in order to keep a balanced study design. Soil samples were collected in spring 2016 and winter wheat (*Triticum aestivum*, L.) sown in autumn 2015 was grown in all fields.

**2.2 Soil sampling**

Soil samples were collected in a circular sampling area of around 300 m$^2$. Wheel tracks were excluded for sampling. Undisturbed cylindrical soil core samples of 100 ml volume and 5.1 cm diameter were taken for different soil physical measurements including gas transport properties and water holding capacity. Composite samples were used for the determination of soil texture, soil organic carbon content, and microbial biomass and respiration. Samples were taken from two different depths representing the topsoil and the subsoil of each field. Undisturbed cylinder samples were sampled from 10-15 cm, and 35-40 cm depth, while composite samples were taken from 5-20 cm and 25-50 cm depth. Hence, the mean sampling depths for both types of samples were 12.5 cm and 37.5 cm (Fig. 1). Five undisturbed cylinder samples were collected per depth and field. The composite samples consisted of 15-20 separate auger samples. Both, undisturbed cylinder and composite samples were taken evenly spaced along two transects crossing the sampling area of 300 m$^2$.

**2.3 Measurements on undisturbed cylinder samples**

The samples were closed at the bottom and the top and stored in the dark at 4 °C until processing. The soil samples were weighed to determine soil moisture at sampling. The soil cylinders were slowly saturated from below, and equilibrated on a ceramic suction plate to 30 hPa and 100 hPa matric suction. By weighing at saturation and at both matric suctions the respective gravimetric water contents were calculated. Gravimetric water content at 100 hPa matric suction, which is typically seen to represent field capacity (Schjonning and Rasmussen, 2000), is defined in this paper as the soil water holding capacity (WHC). The relative gas diffusion coefficient and air permeability were measured at 30 hPa and 100 hPa matric suction as described by Martínez et al., (2016a). To obtain soil bulk density the soil samples were dried at 105 °C for at least 72 h before weighing. Volumetric water content was calculated from bulk density and gravimetric water content. Total porosity was determined based on soil bulk density and particle density, which was measured separately for each field and sampling depth. Finally, air-filled porosity ($\varepsilon_a$) at 30 hPa and 100 hPa matric suction was calculated from total porosity and the respective volumetric water content.



## 2.4 Measurements on composite samples

Composite samples were processed following the reference method of the Swiss Agricultural Research Stations (Swiss Federal Research Stations, 1996). Before measurements were performed, the composite samples were cleaned from animal and plant debris and then sieved at a mesh width of 2 mm. Soil texture was determined with the pipette method, while soil organic carbon was quantified by the dry combustion method according to ISO 10694. Soil microbial biomass was estimated from substrate induced respiration measurements as described by Anderson and Domsch, (1978). An equivalent of 50 g soil dry matter was amended with 150 mg glucose before incubation at 22° C for seven days. Following Heinemeyer et al., (1989), an infrared gas analyser was used to measure initial $CO_2$ release. Soil microbial biomass (Cmic) was then calculated from these initial respiration rates according to Kaiser et al., (1992) assuming 1 µl $CO_2$ g$^{-1}$ dry soil h$^{-1}$ to be equivalent to 30 µg Cmic g$^{-1}$ dry soil. Furthermore, soil basal respiration (microbial respiration) was measured in pre-incubated samples (7 days at 22 °C, equivalent of 20 g dry soil) as $CO_2$ released over a 48 h period, starting at the second day of the incubation (Swiss Federal Research Stations, 1996).

## 2.5 Soil penetration resistance

Additional soil physical information was obtained from cone penetrometer measurements. Ten separate penetrometer insertions were performed in each field across the 300 m$^2$ sampling area down to a depth of 50 cm (Fig. 1) using an Eijkelkamp penetrologger (Eijkelkamp Agrisearch Equipment, Giesbeek, The Netherlands). The cone had a base are of 1 cm$^2$ and a full apex angle of 60°, and penetration resistance values were obtained in 1 cm steps. To represent the same soil depths as for composite and cylinder samples, average values were calculated for 5 cm to 20 cm depth and 25 to 50 cm depth. Soil water content at the time of penetrometer measurements was obtained from the undisturbed soil cylinder samples.

## 2.6 Data analysis and statistics

Data analysis and statistics were performed in R version 3.4.1 (R Core Team, 2017). The effects of soil management and sampling depth on the different soil properties were evaluated with linear mixed models using the nlme package (Pinheiro et al., 2013). The following model, which was followed by analysis of variance (ANOVA), was used to evaluate whether soil texture significantly differed among management systems and sampling depths:

$$Y_{ijk} = \alpha_i + \beta_j + \alpha\beta_{ij} + \gamma_k + \varepsilon_{ijk} \,, \tag{1}$$

where $Y$ represents the clay, slit or sand content in the i$^{th}$ management treatment (i = conventional, no-till, organic) of the j$^{th}$ depth (j = 12.5 cm, 37.5 cm) and the k$^{th}$ field (k = 1, 2,…, 29, 30); $\alpha$ denotes the effect of the management treatment, $\beta$ denotes the effect of the depth, $\alpha\beta$ represents the interaction between management and depth and $\varepsilon$ represents the residual error. The effects of the management system, the depth and their interaction were treated as fixed effects. To account for possible autocorrelation of soil properties taken at different depths in the same field, the effect of the field ($\gamma$) was included into the



model as a random factor. The following model was used to evaluate whether total and air-filled porosity, gas transport properties, water holding capacity, soil organic carbon content, microbial biomass and respiration, respectively, were affected by the management system and the soil depth:

$$Y_{ijk} = \alpha_i + \beta_j + \alpha\beta_{ij} + \gamma_k + \delta + \varepsilon_{ijk} \,, \tag{2}$$

As in Eq 1, the effects of soil management, depth and their interaction are denoted by $\alpha$, $\beta$ and $\alpha\beta$, respectively, and were set as fixed effects while the field ($\gamma$) was set as a random effect. The effects of clay content ($\delta$) was included as a fixed co-variable into the model. This allowed to account for the variability of soil texture among sites and thus to account for effects related to the site-specific soil texture. For soil penetration resistance, the gravimetric water content at sampling was added as an additional fixed co-variable to the model, due to known influence of soil moisture on penetration resistance (Bengough et al.,

2011; Busscher, 1990). Analysis of covariance (ANCOVA) was then used to test significant effects of fixed factors. Air permeability was transformed to base 10 logarithm for linear mixed model analysis. Pairwise comparison of group mean values within one sampled depth were performed using least significant difference tests (LSD) at $p < 0.01$, $p < 0.05$ and $p < 0.1$, respectively, using the "agricolae" package for R (Mendiburu, 2015).

The following multiple linear regression model was applied to explain soil organic carbon content as a function of soil physical

properties:

$$SOC \,[g\ C\ kg^{-1}\ soil] = a * x_1 + b * Clay\ [\%] + c \,, \tag{3}$$

where *SOC* represents soil organic carbon content and the first explanatory variable ($x_1$) represents either gas diffusivity, air permeability, air-filled porosity, soil penetration resistance or water holding capacity. As done in Eq. 2, clay content was included as a second explanatory variable into the regression model, which allowed to account for the site specific soil textural

composition. The same regression model was used to explain microbial biomass as a function of gas diffusivity, air permeability, air-filled porosity ($x_1$) and clay content. Effects of soil organic carbon content on soil physical properties were evaluated with the following regression model:

$$Y = a * SOC \,[g\ C\ kg^{-1}\ soil] + b * Clay\ [\%] + c \,, \tag{4}$$

where *Y* represents gas diffusivity, air permeability, air-filled porosity, soil penetration resistance or water holding capacity.

As described above, clay content was included into the model to account for the textural variability among sites. Due to strong effects of the sampling depth on all soil properties assessed, all multiple linear regressions were performed separately for each soil depth. Furthermore, regressions were carried out separately for both levels of matric suction at which gas diffusivity, air permeability and air-filled porosity were determined. Simple linear regressions were used to relate soil organic carbon content, microbial biomass and respiration. All regression models were evaluated with the linear least square method (lm), which is

implemented into the R package "stats" (R Core Team, 2017).





## 3 Results

Considerable variability in soil textural composition was observed among the 30 fields that were part of the study. Clay, silt and sand contents varied from 11.0% to 48.4%, 19.7% to 43.9% and 16.5% to 60.4%, respectively. Soil texture did not differ significantly between the three different soil management systems. Slightly lower clay and higher sand contents were observed
in the topsoil in comparison to the subsoil (Table 1).

### 3.1 Effects of management system on soil physical properties

Total porosity in the topsoil was significantly lower in the no-till system than in the two management systems that include regular tillage (p < 0.01). In the no-till system, mean total porosity at 12.5 cm depth was 48.6% (± 3.1% standard deviation (SD)). Total topsoil porosity of the conventional and organic management system was 54.5% (± 3.6% SD) and 56.1% (± 4.8%
SD), respectively (Fig. 2). Consequently, soil bulk density at 12.5 cm depth was higher in the no-till system than in the two management systems that include regular tillage. No significant difference in total porosity and bulk density was observed between the management systems in the subsoil, i.e. at 37.5 cm depth (Fig. 2).

Air-filled porosity in the topsoil at both matric suctions was significantly higher (LSD test: $p < 0.01$) in the organic and conventional system compared to the no-till system (Supplemental Fig. S1). Similar effects of the management system
occurred for gas transport capability of the soil, i.e. relative gas diffusion coefficients and air permeability. Compared to the no-till system, gas diffusion coefficients at 100 hPa matric suction were higher by more than 70% in the organic and conventional system (least significant difference (LSD) test: $p < 0.05$). Similar differences between management systems were observed for air permeability at 100 hPa, which was significantly higher (LSD test: $p < 0.01$) in the organic and conventional systems than in the no-till system (Fig. 2). At matric suction of 30 hPa similar but less pronounced differences of gas transport
properties between management systems occurred as under drier conditions at 100 hPa matric suction (Supplemental Fig. S1). In the subsoil however, gas diffusion coefficients, air permeability and air-filled porosity did not differ significantly between the three management systems (Fig. 2 and Supplemental Fig. S1). Water holding capacity was higher in the organically managed system than in the no-till system in both the topsoil and the subsoil (LSD test: $p < 0.01$). In the conventional management system, water holding capacity was in between the organic and no-till system (Fig. 2). Soil penetration resistance
also significantly differed between the management systems. Mean penetration resistance between 5 cm and 20 cm depth was more than 35% lower in the organically and conventionally managed system than in the no-till system. In the subsoil, i.e. between 25 and 50 cm depth, around 20% higher penetration resistance was observed in the no-till system compared to the systems that were conventionally and organically managed. However, these differences were only significant at the 10% significance level. Most likely because of the relatively low variation in soil moisture at sampling, no significant (p = 0.28)
effects of soil moisture on penetration resistance was found (Fig. 3). Despite significant effects of the management system, considerable overlap between the systems was found for all assessed soil physical properties (Fig. 2 and 3, and Supplemental Figure S1).

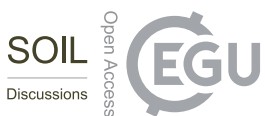

Significant effects of the soil depth were observed for all soil physical properties. Total porosity, gas diffusion coefficient, air permeability and water holding capacity, as well as air-filled porosity (Fig. 2; Supplemental Fig. S1) were lower in the subsoil than the topsoil. Penetration resistance was higher in the subsoil than in the topsoil (Fig. 3). Furthermore, clay content and thus the site-specific soil texture affected total and air-filled porosity, air permeability, and water holding capacity. For gas diffusion

coefficients and penetration resistance, no significant effect of clay content ($p > 0.05$) was observed (Fig. 2 and 3; Supplemental Table S1 and Supplemental Fig. S1).

### 3.2. Effects of management system on soil organic carbon content, microbial biomass and respiration

Similar to the results obtained for soil physical properties, significant effects of the management system on soil organic carbon content and microbial biomass were observed. In the topsoil and the subsoil the highest organic carbon content and microbial

biomass was found in the organic system (Table 2). Microbial respiration did not differ significantly among the three management systems. Soil organic carbon content as well as microbial biomass and respiration significantly decreased with soil depth (Table 2) and were affected by the clay content (Supplemental Table S2). Similar to the results obtained for soil physical properties, considerable overlap between the management systems was also found for organic carbon content, microbial biomass and respiration (Table 2).

### 3.3. Interrelations between soil physical properties and soil organic carbon content

Multiple linear regression models (Eq. 3) were used to relate soil physical properties to soil organic carbon content. Soil organic carbon content significantly increased with increasing gas diffusion coefficients and air permeability. For the topsoil, multiple $R^2$ values were in the range 0.60 to 0.68, and the regression coefficients for gas diffusivity and air permeability measured at 100 hPa matric suction were highly significant ($p < 0.01$). Also in the subsoil increased gas diffusivity ($p < 0.05$, $R^2 = 0.53$)

and air permeability ($p < 0.01$, $R^2 = 0.65$) resulted in higher soil organic carbon content (Fig. 4). Similar results were obtained when using gas transport properties measured at 30 hPa instead of the transport properties obtained at 100 hPa to explain organic carbon content (Supplemental Fig. S2). In contrast, no clear relationship between air-filled porosity and soil organic carbon content was observed since the regression coefficients for air-filled porosity were mostly not significant (Supplemental Fig. S3). Increased water holding capacity was significantly ($p < 0.01$) related to higher soil organic carbon content in both

soil layers ($0.57 < R^2 < 0.64$; Fig. 5). Despite trends towards lower soil organic carbon contents with increased soil penetration resistance in the topsoil, no strong relationships between penetration resistance and soil organic carbon content were found (Table 3). Across sampling depths and at both levels of matric suction, clay content was positively ($p < 0.01$) related to soil organic carbon content (Table 3, Fig. 4, Supplemental Fig. S2 and S3).

Using a second set of regression models (Eq. 4) showed that soil physical conditions improved with increasing organic carbon

content. Gas diffusivity (topsoil: $p < 0.05$, subsoil: $p < 0.01$) and air permeability (topsoil and subsoil: $p < 0.01$), significantly increased with increasing soil organic carbon content. The coefficients of determination were between 0.23 and 0.50 (Supplemental Table S3) and thus considerably lower than the $R^2$ values obtained from Eq. 3 (Fig. 4 and Supplemental Fig.





S2). No strong influence of soil organic carbon content on air-filled porosity was observed ($0.10 < R^2 < 0.29$, Supplemental Table S3). Water holding capacity significantly increased with soil organic carbon content ($p < 0.01$; $0.73 < R^2 < 0.75$). No significant impact of organic carbon content on penetration resistance was found (Supplemental Table S4).

The results obtained from the regressions analyses (Eq. 3 and 4) show that soil organic carbon content is closely interrelated with soil physical properties, namely gas transport capability and water holding capacity. Soil organic carbon content was positively influenced by improved soil physical conditions, while in turn, water holding capacity and gas transport properties were positively affected by soil organic carbon content (Fig. 6).

### 3.4. Effects of gas transport properties and soil organic carbon content on microbial biomass and respiration

Soil microbial biomass could also be explained as a as a function of gas transport properties and clay content (Eq. 3), especially in the topsoil. As observed for soil organic carbon content, higher clay content significantly ($p < 0.01$) increased microbial biomass in both soil layers (Table 4, Supplemental Table S5). In the topsoil increased gas diffusivity ($p < 0.01$, $0.57 = R^2$) and air permeability ($p < 0.05$, $R^2 = 0.50$) as measured at 100 hPa matric suction resulted in higher microbial biomass (Table 4). At 30 hPa matric suction similar relationships between gas transport capability and microbial biomass were observed (Supplemental Table S5). Furthermore, increased air-filled porosity resulted in higher microbial biomass in the topsoil (Supplemental Table S6). In the subsoil however, no clear relationship between soil microbial biomass and gas transport properties (Table 4, Supplemental Table S5) and air-filled porosity (Supplemental Table S6), respectively, were observed. Microbial biomass was also positively correlated with soil organic carbon content both in the topsoil and the subsoil ($p < 0.01$; $R^2 = 0.75$; Table 5). In both soil layers, increased microbial biomass was significantly ($p < 0.01$) correlated with higher microbial respiration (topsoil: $R^2 = 0.83$; subsoil: $R^2 = 0.84$; Table 5).

### 4 Discussion

Here, we investigated the interrelations between soil physical properties and soil organic carbon content in arable soils. Other than in most field plot studies where different management treatments are compared on the same soil, the current on-farm study included 30 fields with substantial variation in soil texture (Table 1). Our results clearly show that soil management at the farm level significantly affects a range of soil properties. Significant differences between the three management systems were measured for most soil properties, despite considerable overlap between the systems. Tillage increased gas transport rates and water holding capacity (Fig. 2), and decreased penetration resistance (Fig. 3) and thus resulted in soil physical conditions that are known to facilitate root growth (Bengough et al., 2011; Rich and Watt, 2013). Our results correspond to findings from field plot experiments, but the effects were restricted to the topsoil, which is in contrast to certain studies (Azooz et al., 1996; Carter et al., 2007; Dal Ferro et al., 2014; Kahlon et al., 2013; Martínez et al., 2016a, 2016b; Pires et al., 2017; Schjonning and Rasmussen, 2000). Further significant differences among the management systems occurred for soil organic carbon content and microbial biomass. As shown previously (Birkhofer et al., 2008; Gattinger et al., 2012; Mäder et al., 2002), organic farming




resulted in increased soil organic carbon content and higher microbial biomass in both soil layers. Other than in previous studies (Carter et al., 2007; Kahlon et al., 2013; Kuhn et al., 2016; Martínez et al., 2016a; da Silva et al., 2014), no significant differences in soil organic carbon content and microbial biomass between the conventional and no-till system were found (Table 2).

Including 30 different fields resulted in large variation of soil texture, gas transport properties and soil organic carbon content. This variation enabled us to show that soil organic carbon content can be explained as a function of soil gas transport capability and clay content (Eq. 3). Soil organic carbon content significantly increased with higher gas transport capability in the topsoil and the subsoil (Fig. 4). Previous studies showed that soil aeration greatly affects root growth (Bengough et al., 2011; Dresbøll et al., 2013; Qi et al., 1994; Thomson et al., 1992; Watkin et al., 1998) and thus soil organic matter input. Despite these results,

effects of soil aeration on soil organic carbon content are rarely investigated. The results obtained here demonstrate that well aerated soils show higher contents of organic carbon. High soil organic carbon content is also known to improve soil structure and soil aeration (Carter et al., 2007; Martínez et al., 2016b; Young et al., 1998). Accordingly, we found that increased soil organic carbon content resulted in higher gas transport capability (Eq. 4; Supplemental Table S3). The explanatory power however was considerably lower than the $R^2$ values obtained when explaining soil organic carbon content as a function of soil

gas transport properties ($0.53 < R^2 < 0.71$ *vs.* $0.23 < R^2 < 0.50$). In addition, high water holding capacity and low soil penetration resistance also facilitate root growth (Bengough et al., 2011; Rich and Watt, 2013) and thus increase inputs of soil organic matter. We observed a strong positive relationship between water holding capacity and soil organic carbon content (Fig. 5; Supplemental Table S4), while soil penetration resistance and organic carbon content were only weakly related (Table 3, Supplemental Table S4). This can most likely be explained by the relatively moist conditions at sampling (Fig. 3) as

relationships between organic carbon content and penetration resistance are stronger under dry conditions (Soane, 1990).

Based on the presented results and existing literature, we propose a positive feedback cycle between soil structure and related soil physical properties like gas transport capability and water holding capacity, root growth and thus organic matter input and soil organic carbon content. It is known that good soil aeration, high water holding capacity and low penetration resistance increase root growth (Bengough et al., 2011; Jin et al., 2013; Rich and Watt, 2013). Since roots represent the major source for

soil organic carbon (Balesdent and Balabane, 1996; Kätterer et al., 2011; Rasse and Smucker, 1998), soil physical conditions that foster root growth will likely lead to increased soil organic carbon content. In turn, higher soil organic carbon content improves soil structure. This results in improved soil aeration and penetrability, and higher water holding capacity, which further facilitates root growth (Fig. 6). Young et al., (1998) discussed these interrelations between soil physical properties, soil life and soil organic carbon. Thereby, soil aeration is of particular importance because it is not only related to root growth but

also to microbial biomass and thus the decomposition of organic matter (Balesdent et al., 2000). In the current study, gas transport capability in soil (Table 4) and soil organic carbon content were positively related to microbial biomass (Table 5). Anaerobic microsites in soil, which are characterized by minimal microbial activity, are known to be important regulators for the stabilisation of organic carbon (Keiluweit et al., 2016, 2017). Due to root respiration and local carbon dioxide accumulation, such anaerobic microsites are likely to form around roots (Koop-Jakobsen et al., 2018). For the build-up of soil organic carbon

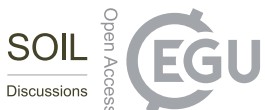

at larger scales however, aerobic conditions are needed as they promote root growth (Dresbøll et al., 2013; Grzesiak et al., 2014; Thomson et al., 1992; Watkin et al., 1998). Furthermore, roots are known to grow towards well aerated soil compartments (Colombi et al., 2017; Porterfield and Musgrave, 1998). Hence, aerobic parts of the soil are likely to be enriched with new organic matter. Given the strong positive relationship between gas transport properties and soil organic carbon

content (Fig. 4), we conclude that in the current study effects of soil aeration on organic matter inputs were more pronounced than stimulation of decomposition.

It has been emphasized that the close interactions between physical, chemical and biological processes need to be accounted for when evaluating the carbon storage potential of arable systems (Qi et al., 1994; Rasmussen et al., 2018; Young et al., 1998). Despite the knowledge about the influence of soil aeration and oxygen concentration in soil air on root growth, the effects of

soil aeration on soil organic carbon content have received only little attention. Unlike water and mineral nutrients, oxygen is not directly acquired by soil inhabiting organisms including plants, soil fauna and microbes. However, oxygen largely regulates the growth and metabolism of these organisms and thus plays a crucial role in carbon cycling. We demonstrate here that gas transport capability of soil plays a key role for the carbon storage in arable soil. The results of this study were obtained on-farm and therefore represent relevant scales and management conditions. As shown here and in previous studies (Albizua et

al., 2015; Azooz et al., 1996; Carter et al., 2007; Dal Ferro et al., 2014; Kahlon et al., 2013; Martínez et al., 2016b; Pires et al., 2017; Schjonning and Rasmussen, 2000), agricultural management affects soil structure and therefore soil aeration. This has ultimate consequences for root growth (Thomson et al., 1992; Watkin et al., 1998; Dresbøll et al., 2013; Grzesiak et al., 2014), which is the major source of organic carbon in soils (Balesdent and Balabane, 1996; Kätterer et al., 2011; Kong and Six, 2010; Rasse et al., 2005). In turn, higher organic carbon content improves soil structure and associated soil physical properties

including gas transport properties. These interactions can be described by a positive feedback cycle as illustrated in Fig. 6. The presented data demonstrates that aeration is a crucial factor for carbon storage in soil. Thus, soil aeration needs to be taken into account when assessing the carbon sequestration potential of arable soils.

## 5 Conclusions

Based on results from obtained from 30 fields of commercial farms, we show that increased soil aeration and water holding

capacity result in higher levels of soil organic carbon. We propose that this is a consequence of increased carbon inputs due to enhanced root growth in soils that are well aerated and retain enough plant available water. Soil aeration is thereby of particular importance as it is closely related to organic matter input as well as to the decomposition rate of organic matter. Increased soil aeration leads to higher levels of oxygen in soil air and thus reduces the risk for soil hypoxia. In turn, root growth and microbial biomass and activity are increased, which are major drivers for carbon dynamics in soil. Our results suggest that the effects of

aeration on carbon inputs were more pronounced than stimulation of carbon decomposition by heterotrophic soil life. However, opposite relationships between soil aeration and organic carbon content may occur in different land use systems or under different climatic conditions. Nevertheless, we clearly showed that aeration plays a crucial yet underestimated role for the





potential of arable soils to act as a terrestrial carbon sink. Future research and policy measures that aim to increase carbon in arable land use systems need to account for the effects of soil aeration on soil carbon dynamics.

**Author contributions**

TK, JS, MvdH, RC, FW and LB conceived the study; FW, MS, and KL conducted the sampling; FW and MS carried out laboratory measurements; Data analysis and interpretation was performed mainly by TC with contribution from FW, TK and LB; TC prepared the manuscript with contribution from all co-authors.

**Competing interests**

The authors declare that they have no conflict of interest.

**Acknowledgements**

We express our sincere gratitude to all farmers for their cooperation and for the possibility to sample in their fields. Diane Bürge, Andrea Bonvicini and Susanne Müller (Agroscope Zurich) are thanked for their help in the laboratory, and Julia Hess, Joel Hofer, Marcel Meyer and Juselin Gurrier (Agroscope Zurich) are thanked for their assistance during field sampling. This study was funded by the Swiss National Science Foundation (project no 406840-161902) within the National Research Programme NRP 68 "Soil as a Resource" (www.nrp68.ch), which is gratefully acknowledged.

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

**Tables**

**Table 1: Effects of management system (M), sampling depth (D) and their interaction (M:D) on soil texture analysed with linear**
**mixed models (Eq. 1) followed by analysis of variance (ANOVA). ** and ° denotes significant effects at p < 0.01 and < 0.1, respectively. Average values for the different depths represent means of three management systems (no-till, conventional, organic, n=10).**

| | ANOVA | | | Average (±SD) | | |
|---|---|---|---|---|---|---|
| **Soil property** | **M** | **D** | **M:D** | **- 12.5** | **- 37.5** | **overall** |
| Clay [%] | p = 0.41 | ** | p = 0.33 | 22.7 (±8.0) | 24.1 (±7.7) | 23.4 (±7.8) |
| Silt [%] | p = 0.70 | p = 0.33 | p = 0.50 | 34.7 (±4.3) | 34.3 (±4.8) | 34.5 (±4.5) |
| Sand [%] | p = 0.37 | ° | p = 0.22 | 42.6 (±9.2) | 41.7 (±10.0) | 42.1 (±9.6) |

Abbreviation: SD = Standard deviation



**Table 2: Effects of management system (M), sampling depth (D), their interaction (M:D) and clay content on soil organic carbon, microbial biomass and microbial respiration analysed with linear mixed models (Eq. 2) followed by analysis of covariance (ANCOVA). \*\*, \* and ° denotes significant effects at p < 0.01, < 0.05 and < 0.1, respectively. Average values for the different depths represent means for no-till (NT), conventional (CON) and organic (ORG) soil management system. Different letters indicate significant differences between management systems at individual depths using least significant difference (LSD) tests at p < 0.05 (n=10).**

| Soil property | ACNOVA | | | | Depth [cm] | Average (±SD) | | | |
| | M | D | M:D | Clay [%] | | NT | CON | ORG | LSD |
| --- | --- | --- | --- | --- | --- | --- | --- | --- | --- |
| SOC [g C kg$^{-1}$ soil] | ° | ** | p = 0.83 | ** | -12.5 | 18.2ab (± 8.5) | 17.3a (± 5.8) | 24.4b (± 13.6) | 0.70 |
| | | | | | -37.5 | 9.4a (± 3.6) | 10.7a (± 4.2) | 16.2b (± 11.4) | 0.53 |
| micC [mg C kg$^{-1}$ soil] | * | ** | p = 0.39 | ** | -12.5 | 607a (± 354) | 573a (± 185) | 861b (± 417) | 247 |
| | | | | | -37.5 | 261a (± 149) | 317ab (± 212) | 467b (± 301) | 170 |
| Resp [µg CO$_2$–C g$^{-1}$ soil h$^{-1}$] | p = 0.21 | ** | p = 0.29 | ** | -12.5 | 0.72 (± 0.44) | 0.60 (± 0.19) | 0.87 (± 0.39) | 0.29 |
| | | | | | -37.5 | 0.35 (± 0.24) | 0.40 (± 0.26) | 0.51 (± 0.36) | 0.20 |

Abbreviations: SD = Standard deviation, SOC = soil organic carbon, micC = soil microbial carbon, Resp = soil microbial respiration





**Table 3: Summary statistics of multiple linear regression models to explain soil organic carbon as a function of penetration resistance and clay content (Eq. 4). ** and ° indicates significant regression coefficients at p < 0.01 and p < 0.1, respectively, ns indicates nonsignificant regression coefficients. $R^2$ represents multiple r-squared.**

| Response variable | Depth [cm] | Z [MPa] | Clay [%] | Int | $R^2$ |
|---|---|---|---|---|---|
| SOC [g C kg$^{-1}$ soil] | -12.5 cm | -3.936° | 0.867** | 5.416 ns | 0.63 |
| | -37.5 cm | -1.655 ns | 0.682** | -0.340 ns | 0.45 |

Abbreviations: SOC = soil organic carbon, Z = soil penetration resistance, Clay = clay content, Int = intercept

**Table 4: Summary statistics of multiple linear regression models to explain microbial biomass as a function of gas diffusivity and clay content, and air permeability and clay content (Eq. 3). Gas diffusivity and air permeability were measured at matric suction of 100 hPa. **, * and ° indicates significant regression coefficients at p < 0.01, p < 0.05 and p < 0.1, respectively, ns indicates nonsignificant regression coefficients. $R^2$ represents multiple r-squared.**

| Response variable | Depth [cm] | Dp/D0 [-] | Clay [%] | Int | $R^2$ |
|---|---|---|---|---|---|
| micC [mg C kg$^{-1}$ soil] | -12.5 cm | 8012** | 31.74** | -303.7° | 0.57 |
| | -37.5 cm | 8934° | 21.62** | -293.4° | 0.47 |
| | | **Ka [µm$^2$]** | **Clay [%]** | **Int** | **$R^2$** |
| | -12.5 cm | 2.208* | 31.30** | -197.1 ns | 0.50 |
| | -37.5 cm | 2.766° | 20.47** | -208.6° | 0.47 |

10 Abbreviations: micC = soil microbial carbon, Dp/D0 = gas diffusion coefficient, Ka = air permeability, Clay = clay content, Int = intercept

**Table 5: Summary statistics of linear regression models to explain microbial biomass and respiration as a function of organic carbon and microbial biomass, respectively. ** indicates significant regression coefficients at p < 0.01, ns indicates nonsignificant regression coefficients. $R^2$ represents multiple r-squared.**

| Regression model | Depth [cm] | a | b | $R^2$ |
|---|---|---|---|---|
| micC [mg C kg$^{-1}$ soil] = a* SOC [g C kg$^{-1}$ soil] + b | -12.5 cm | 30.00** | 81.10 ns | 0.75 |
| | -37.5 cm | 26.93** | 24.02 ns | 0.75 |
| Resp [µg CO$_2$–C g$^{-1}$ soil h$^{-1}$] = a* micC [mg C kg$^{-1}$ soil] + b | -12.5 cm | 0.001** | 0.095 ns | 0.83 |
| | -37.5 cm | 0.001** | 0.033 ns | 0.84 |

Abbreviations: micC = soil microbial carbon, SOC = soil organic carbon, Resp = microbial respiration



**Figures**

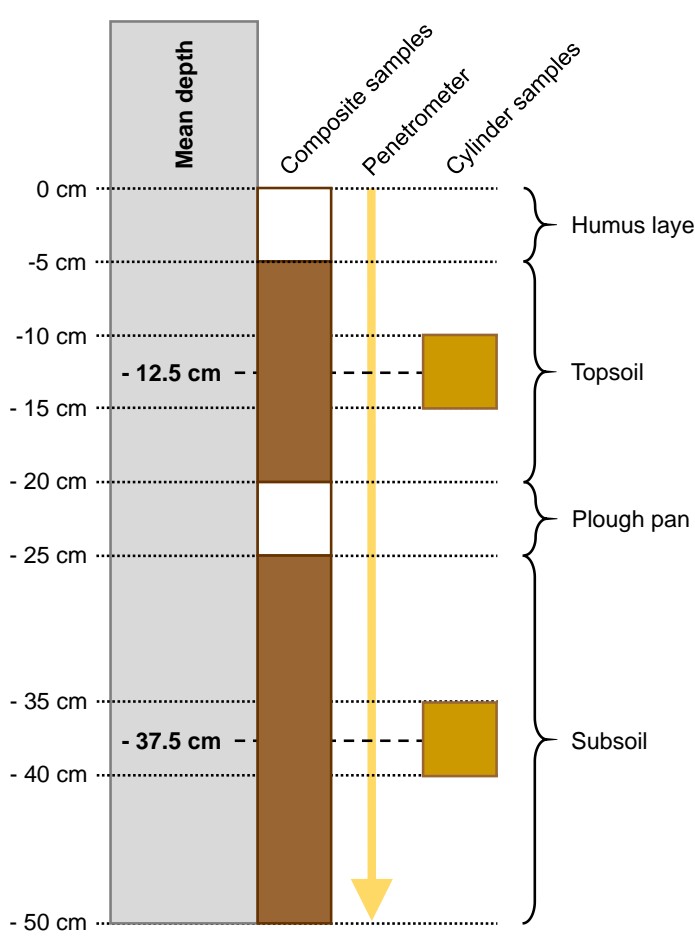

**Figure 1: Schematic overview indicating soil layers and respective depths used for composite samples, penetrometer insertions and cylindrical soil core samples.**



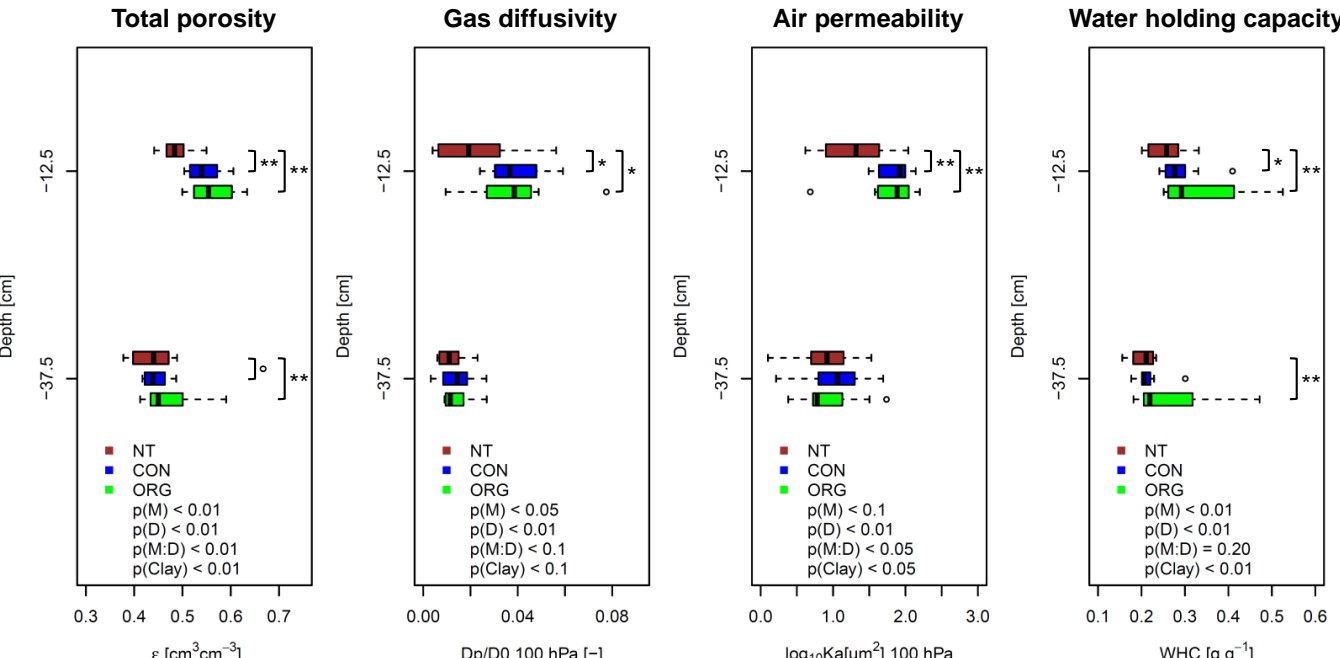

**Figure 2: Effects of soil management (M), sampling depth (D), their interaction (M:D) and clay content (Clay) on total soil porosity, gas diffusivity and air permeability at 100 hPa matric suction and water holding capacity analysed with linear mixed models (Eq. 2) followed by analysis of covariance. NT (red), CON (blue) and ORG (green) denote no-till, conventional and organic management system, respectively. \*\*, \* and ° indicate significant differences**
5 **between management systems at individual depths using least significant difference tests at p < 0.01, p < 0.05 and p < 0.1, respectively (n = 10).**





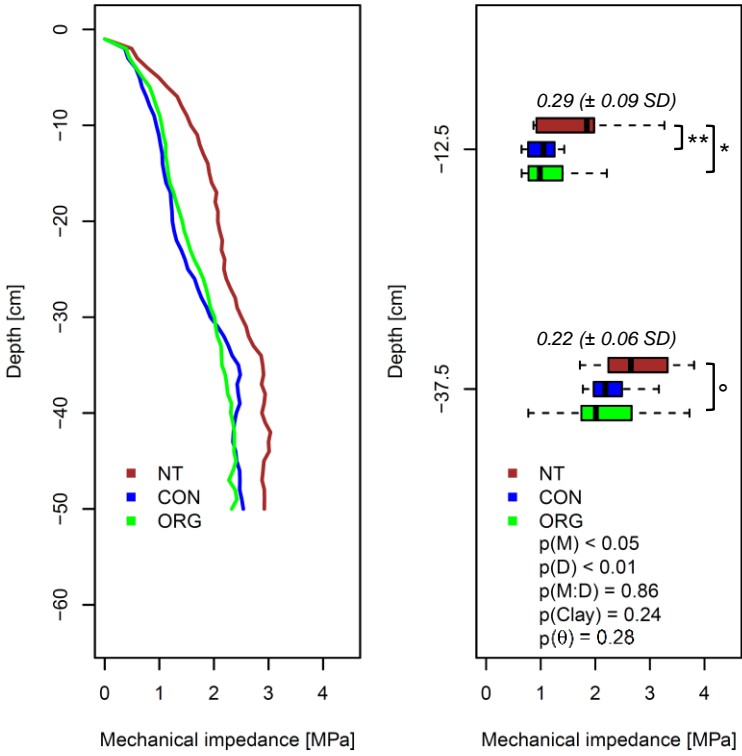

**Figure 3: Effects of soil management (M), sampling depth (D), their interaction (M:D), clay content (Clay) and gravimetric water content at sampling (θ) on soil penetration resistance analysed with linear mixed models (Eq. 2) followed by analysis of covariance. NT (red), CON (blue) and ORG (green) denote no-till, conventional and organic management system, respectively. \*\*, \* and °**
5 **indicate significant differences between management systems at individual depths using least significant difference tests at p < 0.01, p < 0.05 and p < 0.1, respectively. Numbers in italic denote overall mean gravimetric water content (± standard deviation) at the time of measurement (n = 10).**



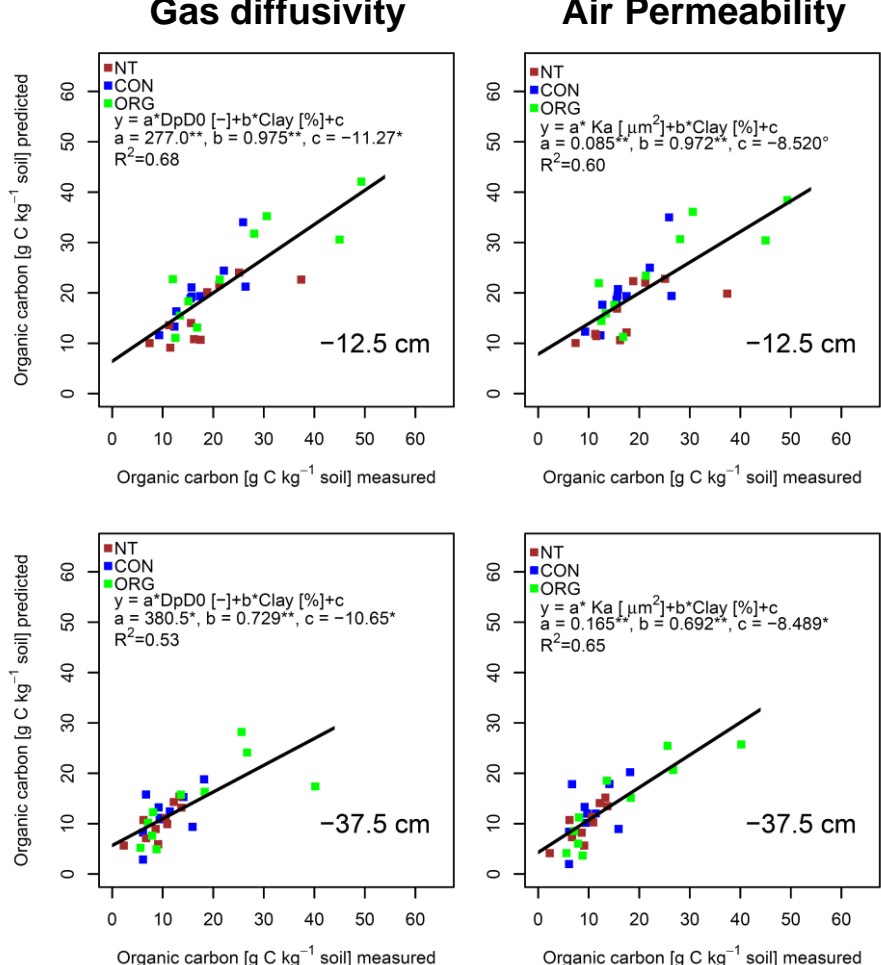

**Figure 4: Multiple linear regression models (Eq. 3) to explain soil organic carbon as a function of gas diffusion coefficients (Dp/D0 [-]) and air permeability (Ka [μm2]) measured at 100 hPa matric suction and clay content (Clay [%]). NT (red), CON (blue) and ORG (green) denote no-till, conventional and organic management system, respectively. \*\*, \* and ° indicates significant regression coefficients at $p < 0.01$, $p < 0.05$ and $p < 0.1$, respectively. $R^2$ represents multiple r-squared.**




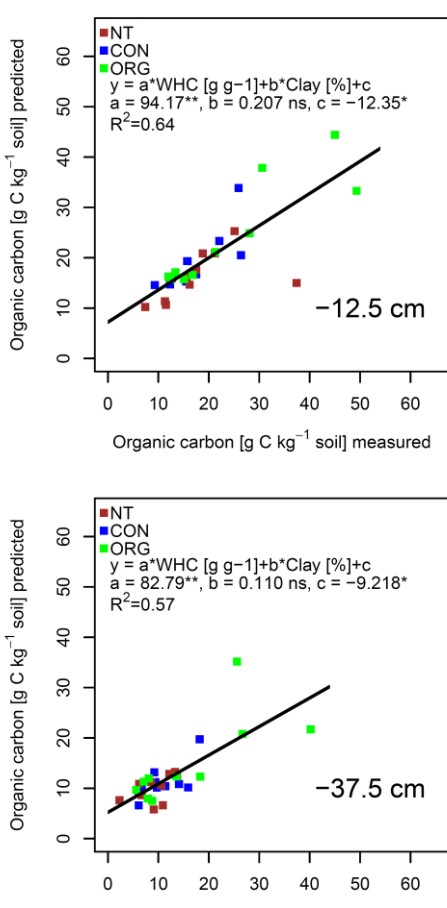

**Figure 5: Multiple linear regression models (Eq. 3) to explain soil organic carbon as a function of water holding capacity (WHC [g g⁻¹]) and clay content (Clay [%]). NT (red), CON (blue) and ORG (green) denote no-till, conventional and organic management system, respectively. ** and * indicates significant regression coefficients at p < 0.01 and p < 0.05, respectively, ns indicates nonsignificant regression coefficients. R² represents multiple r-squared.**

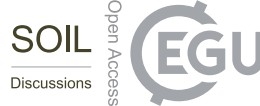

**Carbon sequestration potential**

**Figure 6: Conceptual model illustrating the influence of soil structure on soil organic carbon content. Improved soil aeration, water holding capacity and soil penetrability leads to better physical conditions for root growth, which fuels soil organic matter input and increases soil organic carbon content. In turn, soil structure and related physical properties are further improved. Improved soil aeration may also fuel microbial growth and activity and thus accelerate soil organic matter decomposition.**