# Peer review of "On-farm study reveals positive relationship between gas transport capacity and organic carbon content in arable soil"

_SOIL, 2018_

## Referee Comment (RC1) · Anonymous Referee #1 · 7 Nov 2018

General comments The authors present an empirical anaylsis of the relation between soil organic carbon contents measured under no-till, conventional and organic farming practice and static soil physical properties. The manuscript is well-structured and written in a concise style. However, there are major methodological concerns. There is no evidence that the differences in SOC are releated to the farming practice. 10 sites were chosen for each farming practice. Since SOC measurements prior to the change in farming practice are not availble it is not clear whether the SOC changed as a result of the farming practice or whether this is just a result of a random selection of 30 sites. This is corroborated by $p < 0.1$ for a significant difference in SOC according to management (table 2). The topic given in the title and the hypothesis stated in the introduction could not be tested with the data set presented in this study since most

relevant factors are not included. The SOC content is in equilibrium between the C inputs (not measured) and the decomposition of C. The decomposition process strongly depends on soil temperature. The second most relevant variable is water stress (often experessed by effective saturation or water-filled pore space WFP, e.g. Manzoni et al., 2012, Ecology 93). Even if the same temperature regime is assumed, the soil water status probably differs considerably between the sites and treatments investigated in this study. Both factors, carbon inputs and soil water status are not included in the data-set presented in this study. The third most relevant factor might be soil aeration. Of course, when the most relveant factors were excluded from the statistical analyses, soil aeration appears to matter... And soil water content at sampling or water holding capacity are surely not sufficient to account for the effect of water stress on microbial SOC decomposition. Soil aeration deficits in agricultural topsoils are very unlikely. Strong effects of oxygen deficits were observed at O2 concetrations < 0.04 cm3/cm3, e.g Glinski  Stepniewski, 1985, Soil aeration and its role for plants. This indicates that the topsoil has to be saturated almost entirely with water, what rarely happens. This is related to very high precipitation events only, and even then in a structured topsoil with macropores oxygen supply could be sufficient due to the diffusion of oxygen into water-saturated aggregates (Hojberg et al., 1994, Soil Sci. Soc. Am. J. Diffusion coefficients in near-saturated, structured soils are high, see Kristensen et al., 2010, J. Contam. Hydrol. 115. Soil aeration is a complex and temporally and spatially highly variable process. In order to test the hypothesis stated in the introduction a data set comprising carbon contents at the beginning and the end of the experiment, time-series of soil temperature, time-series of soil water contents, time-series of soil CO2 concentrations, time-sereis of O2 concentrations or at least redox potentials and time-series of soil heterotrophic respiration are required. I suggest to re-analyse the present dataset with a new focus.

Specific comments p2 25 there are approaches that account for macropore tortuosity and thir effects on gas diffusion, e.g. Kristensen et al., 2010, J. Contam. Hydrol. p2 29-31 yes, but this means that increaed aeration will lead to lower SOC, since decomposition rates would be higher. This is a clear contradiction to what is hypothesized in the abstract: '...that improved soil aeration, which is strongly controlled by soil structure, leads to higher soil organic carbon content.' p3 13-15 I strongly disagree. There is a bunch of literature (actually an entire community) that found soil temperature and secondly soil water content to be the most relvant drivers of carbon turnover in soils. p3 23 but how will you separate the confounding effects of increased aeration and limited water ability for decomposition? Both are highly inter-related. Low water contents, leading to decreased SOC decomposition, are inherently linked to increased soil aeration and vice versa. p3 16-28 a clear mechanistic description of the processes and the status variables that affect soil aeration and its consequences on SOC decomposition is missing p4 1-2 do you really expect measurable and significant differences in SOC after 5 years? There is a clear lack of data on C inputs. p6 1-2 water holding capacity is not a good proxy for he dynamics of soil water content or water-filled pore space p8 13 exacly, there is considerable overlap, which also causes a rather higher error probablity ($p<0.1$) ... p8 24-15 What would be the effect of increased porosity/water holding capacity? The same amount of water (precipitation) infiltrating into a larger volume will cause less water-filled porosity. This in turn will cause less SOC decomposition, which subsequently leads to higher SOC contents. I suspect a spurious relationship between air permebility/gas diffusivity and SOC. I assume the true correlation is between water-filled porosity and SOC content. p10 9-10 I strongly disagree. The effect of soil aeration on SOC decomposition is well documented in literature. This is text book knowledge, see Glinski Stepniewski, 1985, Soil aeration and its role for plants, chapter I, section II, A.4 and A.5 p11 5-6 This is highly deculative. Neither organic matter inputs nor the main drivers of decomposition were included in the data-set. p11 28-30 This statement is probably one of the main conclusions of this study. However, neither carbon inputs nor the stimulation of carbon decomposition was measured in this study. This conclusion is specultive at this point and not related to results presented in this study.

---

## Referee Comment (RC2) · Anonymous Referee #2 · 10 Nov 2018

All in all, the present study could be a valuable contribution worth being published in soil, provided that multi-collinearity is taken into account and that relations between SOC or Cmic and soil physical parameters are also looked at for the individual management groups. A revised discussion needs to be more conservative. Ideally also more information is provided on C-inputs. The present conclusion heavily depends on the assumption that root C-inputs were lifted by better soil physical traits. But no proof is provided. Also no proof is given that root-C in these systems forms the single most important precursor of SOC. The consequence is that many of the proposed causal relationships are really tentative. This uncertainty needs to be better echoed into the discussion and title. Having said that, the topic is really pertinent, all texts are well written and results have been well presented, and all is based on an impressive volume of

work.

The introduction in principle reads well but should be condensed a bit further. To my impression the article really starts with p1L16. The preceding part could in fact be entirely omitted. It would also be better to place the research hypothesis (p3L23) closer to mention of mechanisms that would explain why better aeration should lead to more C in soil.

M&M P4L9 samples were collected in spring: across a single whole season soil environmental conditions could have evolved: e.g. in April drought and in late May a wet period. As a consequence variables like microbial biomass, respiration and soil penetration resistance are also depending on time of sampling. Was the impact of actual sampling date investigated on the studied soil parameters? Or is this effect negligible? I could assume so for microbial biomass given that the substrate-induced respiration method was used. In other words, was the effect of a covariate sample timing required in the ANOVAs? The statistical approach is very clearly described: really helpful. The approach to cover a wide range in soil texture complicates the analysis but on the other hand promotes representativeness of this work. The authors have done well in accounting for variation caused merely by soil texture. The outcome of the ANOVA is also well presented in the figures. One question though: In the ANOVA of non-texture variables a factor clay content was included, but interactions with the other factors were disregarded. On what basis did you decide to do so? Could we assume the impact of clay% on SOC, Cmic etc. is independent from depth and management? Also, why clay and not silt or sand%? Please comment. I am missing the point why significance of Eq. 4 is introduced as well. Any relations between SOC content and soil physical variables were already investigated by Eq3 . The regression analysis with soil physical traits as dependent variable are apparently redundant.

Results: Results have been well presented, just one remark: L11-12 seems to be in contradiction with Fig. 2: total porosity of the subsoil did significantly differ between management systems. Otherwise a well written and clear section. Generally though,

no account was made of multi-collinearity among explored predictor variables. In 3.4 the positive relation between clay content, air permeability and gas diffusivity and microbial biomass was discussed. But at the same time we know that SOC content dominantly explains 'micC' and 'Resp' (Table 5). It cannot be excluded that mutual positive correlations exist between 1 SOC and 2 clay content, 3 air permeability or gas diffusivity. From the regression models presented in Table 4 it is then not possible to conclude that air permeability or gas diffusivity have a direct significant impact on micC. Their relation may very well be indirectly manifested through SOC content. More conservative regression models that exclude redundant variables are needed here. In any case the authors should more carefully draw conclusions in this study and leave room for alternative explanations than the currently forwarded main conclusion.

Discussion: The current study's main conclusion, viz. aeration, here represented by gas diffusivity and air permeability, significantly controls SOC levels in soils seems premature. This conclusion is drawn from positive linear relations between SOC level an these soil physical variables based on a set of organic and conventionally managed agricultural fields. SOC levels were significantly larger in topsoil of the fields under organic management vs. under conventional management, as could be expected (only organic nutrient sources, cover crops, ley). Obviously a larger SOC level also increases soil strength and lowers soil bulk density, with then also improved aeration. It is then not warranted to immediately conclude that vice versa improved aeration leads to higher SOC levels. Such could only be said if inputs of exogenous OM was more or less constant (aside C inputs from roots) in the investigates set of fields. Separate regressions for on the one hand CT and NT fields and OR fields on the other also need to be presented. If similar trends are found as in the full (n=30) set then indeed the conclusion seems viable. Looking at Fig. 4, with 4 of the OR fields with highest SOC levels, this may not be the case.

The assumption that root-derived C-inputs are superior sources of native SOC in comparison to above-ground plant parts has indeed been demonstrated by several researches. Indeed the so-termed 'relative-contribution factor' of root-C is 2-3x that of above-ground plant parts. But to evaluate if indeed roots form the dominant source of native SOC in the studied fields, the readers need to get more insight into the crop rotations and exogenous OM input management. If in case of OR, exogenous OC input by far exceeds that from roots (more than a factor 2-3) than it seems much less likely that any relation could exist between SOC level and gas diffusivity and air permeability. No root biomass data were supplied to back p11 L5s sub conclusion.

At the same time the authors best recognize that at present also other views exist: several recent studies have highlighted that the aboveground residues are more important for long-term SOM stabilization (HF-SOM) as compared to belowground. This has been often linked with the relatively high decomposability of aboveground residues which generate more microbial by-products, which are actually the precursor of the long-term stabilized SOM, associated with HF e.g. Cotrufo et al., 2013 (GCB), 2015 (nature geosci); Lavallee et al., 2018 (BG).

---

## Author Response (AR1)

**Responses to Reviewers' Comments (manuscript # soil-2018-35)**

**"On-farm study reveals positive relationship between gas transport capacity and organic carbon content in arable soil"**

Tino Colombi, Florian Walder, Lucie Büchi, Marlies Sommer, Kexing Liu, Johan Six, Marcel G. A. van der Heijden, Raphaël Charles and Thomas Keller

Dear Editor,

We herewith submit the response to the comments by the Editor and the two Reviewers to our manuscript (soil-2018-35). First we address the points raised by the Editor. Following that, we provide point-by-point answers to the comments and suggestions provided by the two Reviewers. Major changes to which we refer to in our responses are marked in yellow in the attached manuscript file.

At this point we would also like to thank the Editor and the two Reviewers for constructive and insightful comments, which helped tremendously to improve the manuscript. We hope that our adjustments are satisfactory and that the manuscript is suitable for publication in SOIL.

Sincerely,
Tino Colombi, Florian Walder, Lucie Büchi, Marlies Sommer, Kexing Liu, Johan Six, Marcel G. A. van der Heijden, Raphaël Charles and Thomas Keller

**Responses to comments by Editor**

**Editor comment:** The authors found a positive correlation between soil properties that are related to soil aeration and the soil organic carbon content. Since soil organic matter plays an important role in soil structure formation and hence on soil physical properties like soil hydraulic conductivity, bulk density, gas diffusivity and permeability, this relation is not unexpected neither really novel. Many studies have already made the link between soil physical properties and soil organic matter. What is however novel is that the authors invert the causality of the relation. Their hypothesis is that an increase in soil aeration in arable soils leads to an increase in soil organic matter, rather than that an increase in soil organic matter leads to a better soil aeration. This inversion is interesting and opposite to other causal relations between soil aeration status and soil organic matter content that go in the other direction and that suggest an accumulation of soil organic carbon when organic matter turn-over is hindered by a lack of oxygen. The authors' argument is that with increasing aeration, not only the respiration of organic matter is enhanced but also the input of fresh organic matter via an increased root development. When growth increases more than respiration, then there will be more accumulation and the equilibrium soil organic matter content will shift to higher levels. The crux is that growth should increase more at higher O2 levels than respiration does. Or opposite, the oxygen levels at which growth starts to decline are higher than the oxygen levels at which respiration starts to decline. The authors included references to studies that show that root growth is already limited at higher oxygen concentrations. It would be important to put that against limiting values that are reported for soil respiration.

*Response: We are aware that the effects of soil aeration on both root growth and microbial respiration are extremely difficult to disentangle. One particular difficulty is that the effects of decreasing soil oxygen concentrations in soil air on root growth (i.e. the susceptibility to decreasing oxygen concentrations) may differ substantially among species and between varieties. The physiological reasons for that are manifold. For example, the initiation of root cortical aerenchyma and the development of barriers avoiding oxygen loss are known to differ between species and varieties. These differences in stress tolerance have ultimate implications for root growth and thus soil organic matter input. Hence, we think it would be very speculative to make a comparison between plants and heterotrophic soil life with regard to limiting oxygen concentration in soil air.*
*Reviewer #1 raised similar questions (e.g. soil organic carbon content is defined by balance between organic matter input and decomposition). We hope that by answering the comments from Reviewer #1 it became clearer that we "suggest" (rather than "prove") that the positive relationship between soil organic carbon content in gas transport capability resulted from increased root growth. Please refer to our responses below for page and line number references of the adjustments made.*

**Editor comment:** The question is therefore whether better aeration corresponds with higher root development. Unfortunately, no root mass data are available. But, data on carbon inputs have been included and indicate that there

is not a large variability in carbon inputs in the different fields. This suggests that larger soil organic matter contents must be due to higher input from roots. A question I have is whether yield data could be included. Higher root biomasses could be related to higher aboveground biomass and a positive relation between biomass production (or yield) and soil organic carbon could support the hypothesis. However, the question still remains whether this positive correlation (if it exists) is due to the effect of soil organic matter on soil aeration. Root growth and biomass could also be affected by other effects of soil organic matter like the release of plant nutrients or a reduction of drought stress by a better water holding capacity of soils with a higher soil organic carbon content. Or, higher nutrient concentrations can be related to a higher clay content, which is also related to a higher soil organic carbon content. Or, higher soil aeration could correspond with more drought stress and lower soil organic degradation. But, this hypothesis would go along with a decrease in plant growth and root carbon input with higher soil aeration. Including alternative hypothesis in the discussion than the main one that is brought forward is therefore important. The issue of several alternative hypotheses goes along with multicollinearity of different parameters that could be responsible for the observed variability of the soil organic carbon content in the different soils. I did not find a clear statement in the authors' responses on how to deal with this issue. Maybe a principal component analyses could be helpful to identify correlated properties. A principal component regression (which by definition does not suffer from multicollinearity) could be useful to identify which (correlated) parameter groups explain the variation in soil organic matter.

*Response 1) **Including above-ground biomass/yield data:** Unfortunately we only have yield data from the year when the sites were sampled (2016). Having data from only one year does not allow us to make reliable statements about possible correlations between productivity and root-derived organic matter input. Furthermore, as the study was carried out on-farm a whole range of crops and/or varieties have been cultivated over the past years. Given this diversity, root-shoot ratios were most likely not constant – both between species (e.g. Hirte et al., 2018, Agric Ecosyst Environ) and between varieties (e.g. Colombi and Walter, 2017, Front Plant Sci). This makes correlations between aboveground plant growth/productivity and organic matter inputs even more difficult and speculative.*

*Response 2) **Alternative hypothesis and collinearity:** We hope that the adjustments made at various points in the text (see responses to reviewer comments for more details) make it now clear that our results suggest (and not prove) that higher soil organic carbon contents resulted from root growth, which was facilitated by higher gas transport capability. Throughout the manuscript we emphasize that we see soil aeration as one of many drivers for soil organic carbon dynamics, not the only or the most important one (pg 1 L19, pg2 L5-L7, pg11 L32-pg12 L1). Furthermore, in the conceptual model shown in Figure 5, we also highlight the fact that higher gas transport capability might fuel microbial activity and thus the decomposition of soil organic carbon. We think that by doing so, alterative hypothesis and interpretations of our results should be adequately represented in the manuscript.*
*We thank the Editor for raising the aspect of collinearity between different predictor variables. Accordingly, we checked for collinearity between the different predictors (clay content and gas transport properties/water holding capacity) used in Eq. 3 (results are displayed in Supplemental Figure S4 and S6) and a clear statement is provided in the response to the comment given by Reviewer #2 (see below).*

**Editor comment:** Furthermore, soil aeration status like soil water status are defined by state variables like oxygen concentration, water content and water potential which vary considerably over time due to the dynamic meteorological boundary conditions and the dynamic respiration. How these state variables change over time and depth in response to external drivers depends on the soil properties. In their study, the authors focus on the soil properties, which influence of course the state variables. But it is important to keep in mind that microbial turnover processes and root groth depend on the state variables (temperature, soil water potential, soil water content, and oxygen concentration) and only indirectly on the physical soil properties

*Response: It is true that we use soil properties that relate to state variables and not state variables per-se in order to make statements about relationships between physical properties and soil organic carbon content. We think that we point out now much clearer that the properties we measured (e.g. gas transport properties) are related to/define potential aeration (pg2 L18-24, pg11 L14-17).*

**Responses to comments by Reviewer #1**

**Reviewer comment:** General comments: The authors present an empirical anaylsis of the relation between soil organic carbon contents measured under no-till, conventional and organic farming practice and static soil physical properties. The manuscript is well-structured and written in a concise style. However, there are major methodological concerns. There is no evidence that the differences in SOC are releated to the farming practice. 10 sites were chosen for each farming practice. Since SOC measurements prior to the change in farming practice are not availble it is not clear

whether the SOC changed as a result of the farming practice or whether this is just a result of a random selection of 30 sites. This is corroborated by p<0.1 for a significant difference in SOC according to management (table 2).

*Response: From this very first comment we draw the conclusion that our main findings were not communicated clear enough in the original submission. We therefore made numerous changes throughout the manuscript (cf. responses to the other comments) in order to better emphasize the aims and major findings/conclusions of our study. In very general terms: The primary aim of the study was not to evaluate whether farming practice affects soil organic carbon content and/or sequestration but to "investigate relationships between gas transport capability and organic carbon content in arable soil covering a range of textural compositions" (cf. last paragraph of the Introduction pg3, L21-27). We chose this focus since soil aeration/gas transport properties is often not included into assessments of soil quality (cf. Figure 2 and Figure 4 in Bühnemann et al., 2018, Soil Biol Biochem), even though it is known that gas transport properties of soil control soil-atmosphere gas exchange and hence oxygen concentration in soil air, which greatly matter for all kinds of biological activity. We address this issue more specifically in the comments below.*

*Further on in the manuscript we present the effects of the different soil management systems on soil organic carbon content as well as soil physical properties including gas transport properties, of which certain were significant (LSD-test at p < 0.05 and < 0.01, Figure 2 and 3, Table 3, Supplemental Figure S1 and S2). However, we now clearly state at multiple occasions that there was considerable overlap between the management systems (pg1 L23-24, pg8 L12-13, L21-23, pg8 L33-pg9 L1, pg10 L25-26) as well as large variation between individual fields for most soil properties assessed (pg7 L18, pg8 L1-2, L23, L32, pg10 L22-25, pg11 L5-6, summarized in Table 1 and Table 2). As the presented study was carried out on-farm, the management systems are likely not as contrasting as in field plot experiments, where typically one or two factors are altered while the rest of the management is the same for the different treatments. Furthermore, variability in soil texture among sites (cf. Table 1), which occurred here but usually does not occur in field plot experiments, showed a significant effect on a range of soil properties (Table 3, Figure 2 and 3, Supplemental Figure S1). Therefore, the effects of soil management on different soil properties (e.g. soil organic carbon content, gas transport capability, etc.) will not be as pronounced as in field plot experiments.*

*Moreover, we did not aim to make statements about carbon sequestration, i.e. comparing organic carbon content before and after conversion from one to another soil management system. In fact, all the investigated farms established their current management more than five years ago (this is why we also state that "The fields were managed following three different management systems: conventional (integrated) farming with no-till practice since at least five years, conventional (integrated) farming with tillage since five or more years, and organic farming with tillage since at least five years", pg3 L30-pg4 L2).*

**Reviewer comment:** The topic given in the title and the hypothesis stated in the introduction could not be tested with the data set presented in this study since most relevant factors are not included. The SOC content is in equilibrium between the C inputs (not measured) and the decomposition of C. The decomposition process strongly depends on soil temperature. The second most relevant variable is water stress (often experessed by effective saturation or water-filled pore space WFP, e.g. Manzoni et al., 2012, Ecology 93). Even if the same temperature regime is assumed, the soil water status probably differs considerably between the sites and treatments investigated in this study. Both factors, carbon inputs and soil water status are not included in the data-set presented in this study. The third most relevant factor might be soil aeration. Of course, when the most relveant factors were excluded from the statistical analyses, soil aeration appears to matter...

*Response: To address this, we rephrased the original hypothesis in a more conservative way that –we think– takes into account the uncertainties associated with our study (pg3, L21-22). We emphasize at multiple occasions the "equilibrium" between carbon inputs and decomposition (pg2, L3-5, pg3 L6-15, pg11 L32-pg12 L7, pg12 L16-17, pg12 L29-pg13 L1, Figure 5). Hence, we acknowledge the important role of both carbon inputs (from amendments, crop residues, aboveground litter and roots) as well as decomposition for soil carbon dynamics. Even more importantly, with the current paper we did not aim to state that soil aeration (and hence gas transport properties) is more important than other phenomena for soil organic carbon content, but to show that it matters. We hope that our adjustments in the text (pg 1 L19-20, pg2 L5-L7, pg11 L32-pg12 L1) could clarify this.*

**Reviewer comment:** And soil water content at sampling or water holding capacity are surely not sufficient to account for the effect of water stress on microbial SOC decomposition. Soil aeration deficits in agricultural topsoils are very unlikely. Strong effects of oxygen deficits were observed at $O_2$ concetrations < 0.04 cm3/cm3, e.g Glinski Stepniewski, 1985, Soil aeration and its role for plants. This indicates that the topsoil has to be saturated almost entirely with water, what rarely happens. This is related to very high precipitation events only, and even then in a structured topsoil with macropores oxygen supply could be sufficient due to the diffusion of oxygen into watersaturated aggregates (Hojberg et al., 1994, Soil Sci. Soc. Am. J. Diffusion coefficients in near-saturated, structured soils are high, see Kristensen et al., 2010, J. Contam. Hydrol. 115. Soil aeration is a complex and temporally and spatially highly variable process. In order to test the hypothesis stated in the introduction a data set comprising carbon contents at the beginning and the end

of the experiment, time-series of soil temperature, time-series of soil water contents, time-series of soil $CO_2$ concentrations, time-sereis of O2 concentrations or at least redox potentials and time-series of soil heterotrophic respiration are required.

*Response: Yes, soil aeration is a highly complex process and is difficult to quantify. This is also why we did not use proxy values (i.e. air-filled porosity, water-filled porosity, degree of saturation) to assess soil aeration but measurements of gas transport properties (i.e. gas diffusivity and air permeability). In doing so, we account for the effects of pore connectivity and pore tortuosity on gas exchange dynamics in soil, which cannot be accounted for when using measurements of air-filled (or water-filled) porosity (pg2 L18-24, and pg9 L18-21).*
*Regarding the comment about limiting $O_2$ concentrations for root growth: Here, we disagree with the comments provided by the Reviewer #1. There are numerous studies from the plant science community (ranging from ecophysiology to agronomy and plant biochemistry), which show that root metabolism undergoes drastic changes and root growth rates decrease at relatively moderated levels of soil hypoxia ($O_2 > 10\%$) and/or very short periods of anoxia (flooding for a couple of hours). Other studies showed that this happens on a regular basis in arable soils. We emphasized on that in the text and provided a number of references (pg. 3 L9-12, pg11 L16-19).*

**Reviewer comment:** I suggest to re-analyse the present dataset with a new focus.

*Response: Following this comment and the comments raised by Reviewer #2, we made numerous changes in the analysis of the data. We now included data on exogenous carbon inputs, both from crop residues and organic amendments, of the last five years (methodology is explained in pg5 L22-31 and pg7 L5-7). In doing so we could show that the amount of exogenous carbon input was similar across the three management systems (pg9 L5-6, Figure 3). Even more importantly, we observed no significant relationship between these inputs and soil organic carbon content (Supplemental Table S9-S13), which is highlighted in the Results (pg10 L5-8), Discussion (pg11 L26-28) and Conclusion (pg12 L26-27) sections. Based on these results we suggest that differences in root growth significantly contributed to differences in soil organic carbon contents among fields (**not between management systems**), (pg10 L11-12 and pg11 L14-24, L26-30).*

Specific comments

**Reviewer comment:** p2 25 there are approaches that account for macropore tortuosity and thir effects on gas diffusion, e.g. Kristensen et al., 2010, J. Contam. Hydrol.

*Response: We agree with that and emphasize on the need to measure gas transport properties directly in the Introduction (pg2 L18-24) and the Results (and pg9 L18-21) in order to account for pore tortuosity and connectivity.*

**Reviewer comment:** p2 29-31 yes, but this means that increaed aeration will lead to lower SOC, since decomposition rates would be higher. This is a clear contradiction to what is hypothesized in the abstract: '...that improved soil aeration, which is strongly controlled by soil structure, leads to higher soil organic carbon content.'

*Response: As emphasized in an earlier comment by Reviewer #1 soil organic carbon content results from the balance between input and decomposition of soil organic matter. With the statement given here (now on pg3 L13-15) we try to make it clear that increased soil aeration might also fuel microbial activity and therefore lead to decomposition of soil organic carbon.*

**Reviewer comment:** p3 13-15 I strongly disagree. There is a bunch of literature (actually an entire community) that found soil temperature and secondly soil water content to be the most relvant drivers of carbon turnover in soils.

*Response: Please refer to our response above, in which we try to make clear that we did not aim to state that aeration is the only, or the most important property regulating carbon cycling in soil, but to show that it matters (pg 1 L19, pg2 L5-L7, pg11 L32-pg12 L1). However, plants represent the only possible source for soil organic carbon besides amendments. Plants are highly responsive to changes in soil oxygen (pg1 L19-20, pg3 L9-13, pg11 L16-19), hence soil aeration is of importance. The same applies of course also for the effects of oxygen concentrations in soil air on microbial growth and activity and thus the decomposition of soil organic carbon (pg1 L19-20, pg3 L13-15, pg11 L32-pg 12 L5, pg12 L29-30). Our empirical data together with existing literature provide evidence that soil aeration is an important factor (but not the only one) for soil carbon cycling.*

**Reviewer comment:** p3 23 but how will you separate the confounding effects of increased aeration and limited water ability for decomposition? Both are highly inter-related. Low water contents, leading to decreased SOC decomposition, are inherently linked to increased soil aeration and vice versa.

*Response: We hope that our rephrased last paragraph of the Introduction addresses this comment (pg3 L21-27). As stated above, we did not aim to play soil aeration as a driver for soil organic carbon content off against other drivers, but to provide empirical and conceptual evidence that soil gas transport properties have an effect.*

**Reviewer comment:** p3 16-28 a clear mechanistic description of the processes and the status variables that affect soil aeration and its consequences on SOC decomposition is missing

*Response: We see this comment related to one of the comments provided by Reviewer #2. Therefore, we adjusted the Introduction in order to make it clearer how soil aeration, biological (both auto- and heterotrophic) activity and soil organic carbon content (both inputs and outputs) are related (pg3 L6-15).*

**Reviewer comment:** p4 1-2 do you really expect measurable and significant differences in SOC after 5 years? There is a clear lack of data on C inputs.

*Response: No we do not and this was not the aim of the study. Please see our responses to the general comments. We completely agree on the need for data on exogenous carbon inputs (was also raised by Reviewer #2) and we included this into the manuscripts as stated above.*

**Reviewer comment:** p6 1-2 water holding capacity is not a good proxy for he dynamics of soil water content or water-filled pore space

*Response: We completely agree with this but we did not use it for that. We used it as an additional soil physical property that has implications for root growth and is associated with soil structure (pg2 L15-18, pg11 L28-32)*

**Reviewer comment:** p8 13 exacly, there is considerable overlap, which also causes a rather higher error probability (p<0.1) ...

*Response: Please refer to the responses to the general comments and our adjustments made that emphasize on the overlap between management systems (pg1 L23-24, pg8 L12-13, L21-23, pg8 L33-pg9 L1, pg10 L25-26). As stated above, we present results from an on-farm study (not experimental field plots) covering a wide range of soil textures, in which contrasts in management between systems might be less pronounced as in filed plot experiments.*

**Reviewer comment:** p8 24-15 What would be the effect of increased porosity/water holding capacity? The same amount of water (precipitation) infiltrating into a larger volume will cause less water-filled porosity. This in turn will cause less SOC decomposition, which subsequently leads to higher SOC contents. I suspect a spurious relationship between air permebility/gas diffusivity and SOC. I assume the true correlation is between waterfilled porosity and SOC content.

*Response: Water-filled porosity (which we see as the complementary of the pore space to air-filled porosity) does not account for pore connectivity/tortuosity (pg2 L19-24). Furthermore, our regression analysis showed that air-filled porosity (and thus water-filled porosity) is less suited as a predictor for soil organic carbon content than gas transport properties (pg9 L18-21, Figure 4, Supplemental Tables S1-S3 and Supplemental Figure S3 and S5).*

**Reviewer comment:** p10 9-10 I strongly disagree. The effect of soil aeration on SOC decomposition is well documented in literature. This is text book knowledge, see Glinski Stepniewski, 1985, Soil aeration and its role for plants, chapter I, section II, A.4 and A.5

*Response: As stated above, numerous previous studies showed that even decreases of just a few percentages of oxygen in soil air might significantly reduce root growth and thus soil organic carbon inputs (pg. 3 L9-12, pg11 L16-17).*

**Reviewer comment:** p11 5-6 This is highly deculative. Neither organic matter inputs nor the main drivers of decomposition were included in the data-set. p11 28-30 This statement is probably one of the main conclusions of this study. However, neither carbon inputs nor the stimulation of carbon decomposition was measured in this study. This conclusion is specultive at this point and not related to results presented in this study.

*Response: We agree that the initial submission of the manuscript contained a number of possibly too speculative statements, which was also remarked by Reviewer #2. Following these suggestions we made numerous changes in order to better address possible uncertainties of our study, i.e. by clearly stating that we propose a positive relationship between gas transport capability and soil organic carbon content, which most likely resulted from increased root growth (pg 1 L26-29, pg 10 L4-8, pg11 L14-19, L21-26, pg 12 L22-25). Furthermore, we clearly state on pg12 L33-pg13 L1 that opposite relationships may occur in different land-use systems and/or climates.*

**Responses to comments by Reviewer #2**

**Reviewer comment:** All in all, the present study could be a valuable contribution worth being published in soil, provided that multi-collinearity is taken into account and that relations between SOC or Cmic and soil physical parameters are also looked at for the individual management groups. A revised discussion needs to be more conservative. Ideally also more information is provided on C-inputs. The present conclusion heavily depends on the assumption that root C-inputs were lifted by better soil physical traits. But no proof is provided. Also no proof is given that root-C in these systems forms the single most important precursor of SOC. The consequence is that many of the proposed causal relationships are really tentative. This uncertainty needs to be better echoed into the discussion and title. Having said that, the topic is really pertinent, all texts are well written and results have been well presented, and all is based on an impressive volume of work.

*Response: We appreciate these critical and highly constructive general comments. In accordance to the suggestion, we changed the title of the manuscript to "On-farm study reveals positive relationship between gas transport capacity and organic carbon content in arable soil". In doing so we believe the mentioned uncertainties (e.g. no data on root biomass) are now better reflected in the title of the manuscript without losing its main finding, which is the positive relationship between soil gas transport capability and soil organic carbon content. Furthermore, we adapted the Abstract following the same line of thoughts, i.e. being more conservative in our conclusions (pg1 L18-30). Regarding the other general comments, please refer to the comments below (i.e. Influence of exogenous inputs of soil organic matter, regressions in individual management groups, collinearity of predictors).*

**Reviewer comment:** The introduction in principle reads well but should be condensed a bit further. To my impression the article really starts with p1L16. The preceding part could in fact be entirely omitted.

*Response: We shortened the first part of the introduction (pg2 L2-12) but did not entirely delete it as we believe that a brief general introduction to organic carbon content of arable soils and its relationship to soil management is needed. Furthermore, we addressed in this very first paragraph some of the comments raised by Reviewer #1.*

**Reviewer comment:** It would also be better to place the research hypothesis (p3L23) closer to mention of mechanisms that would explain why better aeration should lead to more C in soil.

*Response: According to the suggestion, we moved the "aims" of the study closer to the explanation of the mechanisms underlying the relationships between soil gas transport properties, root growth and soil organic carbon content (pg3 L6-27).*

**Reviewer comment:** M&M P4L9 samples were collected in spring: across a single whole season soil environmental conditions could have evolved: e.g. in April drought and in late May a wet period. As a consequence variables like microbial biomass, respiration and soil penetration resistance are also depending on time of sampling. Was the impact of actual sampling date investigated on the studied soil parameters? Or is this effect negligible? I could assume so for

microbial biomass given that the substrate-induced respiration method was used. In other words, was the effect of a covariate sample timing required in the ANOVAs?

*Response: The soil samples were collected within a period of ~40 days between late April and the end of May (information is now included in the Material and Methods section, pg4 L8). In the study we followed a specific sampling design: The different fields were allocated as triplets, which means that one field of each management system (conventional, no-till and organic) were geographically close to each other. Sampling was done following this layout, i.e. one triplet was sampled per day (information is now included in the Material and Methods section, pg4 L9-11). This was done to avoid confounding effects of location/sampling date on the differences between management systems. However, due to this design we are unfortunately not able to disentangle effects of the location from effects of the sampling time with the presented statistical approach (linear mixed model followed by ANCOVA). Nevertheless, we introduced sampling time as an additional predictor variable into the multiple-linear regression models (Eq. 3, pg7 L5-7). This analysis showed that sampling date had no significant influence on soil organic carbon content (pg10 L4-5, Supplemental Tables S5-S8).*

**Reviewer comment:** The statistical approach is very clearly described: really helpful. The approach to cover a wide range in soil texture complicates the analysis but on the other hand promotes representativeness of this work. The authors have done well in accounting for variation caused merely by soil texture. The outcome of the ANOVA is also well presented in the figures. One question though: In the ANOVA of non-texture variables a factor clay content was included, but interactions with the other factors were disregarded. On what basis did you decide to do so? Could we assume the impact of clay% on SOC, Cmic etc. is independent from depth and management?

*Response: This choice was made due to the absence of significant management-depth interactions on clay content (Table 1). A statement addressing this is now included in the Material and Methods section (pg6 L21-23).*

**Reviewer comment:** Also, why clay and not silt or sand%? Please comment.

*Response: We chose clay content because it is commonly seen as the textural fraction that is most closely associated with soil structure and soil physical properties as well as with soil organic carbon content. In addition, clay was suitable as it showed the highest variability (expressed as coefficient of variation) among sites. Two sentences motivating this choice are now included in the Material and Methods section (pg6 L19-21).*

**Reviewer comment:** I am missing the point why significance of Eq. 4 is introduced as well. Any relations between SOC content and soil physical variables were already investigated by Eq3. The regression analysis with soil physical traits as dependent variable are apparently redundant.

*Response: As suggested we have removed Eq. 4 and all associated statements in the Results and Discussion section.*

**Reviewer comment:** Results: Results have been well presented, just one remark: L11-12 seems to be in contradiction with Fig. 2: total porosity of the subsoil did significantly differ between management systems.

*Response: Thank you for this. We have changed the respective statement (pg7 L29-30).*

**Reviewer comment:** Otherwise a well written and clear section. Generally though, no account was made of multi-collinearity among explored predictor variables. In 3.4 the positive relation between clay content, air permeability and gas diffusivity and microbial biomass was discussed. But at the same time we know that SOC content dominantly explains 'micC' and 'Resp' (Table 5). It cannot be excluded that mutual positive correlations exist between 1 SOC and 2 clay content, 3 air permeability or gas diffusivity. From the regression models presented in Table 4 it is then not possible to conclude that air permeability or gas diffusivity have a direct significant impact on micC. Their relation may very well be indirectly manifested through SOC content. More conservative regression models that exclude redundant variables are needed here. In any case the authors should more carefully draw conclusions in this study and leave room for alternative explanations than the currently forwarded main conclusion.

*Response 1) **Collinearity in multiple linear regression models:** We performed linear regressions between the different predictor variables in the multilinear regressions models (i.e. clay content and gas diffusivity/air permeability/water holding capacity, cf. Eq 3) in order to check for possible collinearity. In brief, we did not find significant correlations*

*between gas transport properties and clay content (0.00 < R² <0.11). Hence, we suggest that there was no collinearity between clay content and gas transport properties (pg9 L17-18, Supplemental Fig. S4). Water holding capacity in contrast was correlated with clay content (0.61 < R² < 0.67). Due to this collinearity we additionally performed simple regressions between water holding capacity and soil organic carbon content. As observed in the multiple regression models (Figure 4), higher water holding capacity was related to increased soil organic carbon (Supplemental Figure S6, pg9 L24-26).*

**Response 2) Relationship between gas transport properties and microbial biomass/respiration:** *We agree with this comment. Therefore, we removed the regressions formerly presented in Table 4 and present more conservative regression models, i.e. microbial carbon/respiration as a function of soil organic carbon content/microbial carbon. This is mentioned in the Material and Methods section (pg7 L8-13, Eq. 4, Table 5). The results are also presented/discussed in a more conservative way in order to emphasize that soil aeration might have an indirect effect on microbial biomass and activity by increasing soil organic carbon content (pg10 L13-18, pg11 L32-pg12 L3).*

**Reviewer comment:** Discussion: The current study's main conclusion, viz. aeration, here represented by gas diffusivity and air permeability, significantly controls SOC levels in soils seems premature. This conclusion is drawn from positive linear relations between SOC level an these soil physical variables based on a set of organic and conventionally managed agricultural fields. SOC levels were significantly larger in topsoil of the fields under organic management vs. under conventional management, as could be expected (only organic nutrient sources, cover crops, ley). Obviously a larger SOC level also increases soil strength and lowers soil bulk density, with then also improved aeration. It is then not warranted to immediately conclude that vice versa improved aeration leads to higher SOC levels. Such could only be said if inputs of exogenous OM was more or less constant (aside C inputs from roots) in the investigates set of fields. Separate regressions for on the one hand CT and NT fields and OR fields on the other also need to be presented. If similar trends are found as in the full (n=30) set then indeed the conclusion seems viable. Looking at Fig. 4, with 4 of the OR fields with highest SOC levels, this may not be the case. The assumption that root-derived C-inputs are superior sources of native SOC in comparison to above-ground plant parts has indeed been demonstrated by several researches. Indeed the so-termed 'relative-contribution factor' of root-C is 2-3x that of above-ground plant parts. But to evaluate if indeed roots form the dominant source of native SOC in the studied fields, the readers need to get more insight into the crop rotations and exogenous OM input management. If in case of OR, exogenous OC input by far exceeds that from roots (more than a factor 2-3) than it seems much less likely that any relation could exist between SOC level and gas diffusivity and air permeability.

**Response 1) Exogenous soil organic carbon inputs vs. inputs from roots:** *We consider this as a very valuable comment. Therefore, we present now data on inputs of soil organic carbon (both crop residues and organic amendments such as slurry, manure and compost) over the last five years before fields were sampled. The calculations are based on information we obtained from the farmers (data was available for 29 of the 30 fields, pg5 L22-31). Interestingly, there was no significant difference (p > 0.50, Figure 3) between the three management systems with regard to the amount of exogenous organic carbon input (pg9 L5-6). To evaluate whether exogenous inputs of organic matter influenced soil organic carbon content, we included the total amount of organic carbon input (i.e. sum of residues and amendments) as an additional predictor into the regression models (pg7 L5-7). The results of these regressions are presented as supplementary tables (Supplemental Tables S9-S13) and in the Results (pg10 L5-7), Discussion (pg11 L26-28) and Conclusion (pg12 L26-29) sections. In summary, we did not find any significant correlation between exogenous inputs of organic matter and soil organic carbon content (topsoil and subsoil). Hence, we propose that the differences in soil organic carbon content were caused by difference in root derived carbon. We elaborate on that in more detail (and in a more conservative way as in the original submission) in the Results (pg10 L10-12), the Discussion (pg11 L26-31) and the Conclusion (pg12 L26-29) section.*

**Response 2) Regressions for individual management groups:** *We agree that such an analysis is of value, especially if the aim is to evaluate whether relationships between soil gas transport properties and soil organic carbon content change with soil management. However, in the current study we did not aim to explore this aspect (pg3 L21-27) but to investigate links between soil aeration and soil organic carbon content per-se. Nevertheless, we performed the following additional regression analyses, which are similar to those suggested by Reviewer #2: Instead of performing the regressions for the organic management system on the one hand and the no-till and conventional on the other hand, we did the analysis for the no-till on the one hand and the conventional and organic on the other hand. We chose to do so, i) since the predictor variables (i.e. gas diffusivity, air permeability, air-filled porosity and water holding capacity) were mainly affected by tillage rather than organic farming practice (pg9 L27-32, pg11, L3-4, Figure 2) and ii) since the amount of exogenous organic carbon input was not higher in the organic system than in the two systems that also receive mineral fertilizer (pg9 L5-6, Figure 3). These separate regressions are summarized in the Results section (pg9 L27-32) and included in the Supplement (Supplemental Tables S1-S4). More importantly, these regressions support our main conclusion of the study ("positive relationship between soil gas transport capability and soil organic carbon content") since this positive relationship also occurred when looking at tilled and untilled fields separately.*

**Reviewer comment:** No root biomass data were supplied to back p11 L5s sub conclusion.

*Response: We unfortunately do not have data on root biomass over the last five years (would be an incredible sampling effort to sample 30 fields at multiple locations for five years). In order to address the comment we rephrased statements on root growth in a way that makes clear that we "propose" or "suggest" a positive relationship between certain soil physical properties, root growth and soil organic carbon content (pg 1 L27-29, pg 10 L4-8, pg11 L26-31, pg12 L9-11, pg12 L26-29).*

**Reviewer comment:** At the same time the authors best recognize that at present also other views exist: several recent studies have highlighted that the aboveground residues are more important for long-term SOM stabilization (HF-SOM) as compared to belowground. This has been often linked with the relatively high decomposability of aboveground residues which generate more microbial by-products, which are actually the precursor of the long-term stabilized SOM, associated with HF e.g. Cotrufo et al., 2013 (GCB), 2015 (nature geosci); Lavallee et al., 2018 (BG).

*Response: Thank you for this remark and the link to the references. We addressed it in the Discussion section (pg11 L 7-12).*

[revised manuscript text omitted]

---

## Author Response (AR2)

**Responses to Reviewers' Comments (manuscript # soil-2018-35)**

**"On-farm study reveals positive relationship between gas transport capacity and organic carbon content in arable soil"**

Tino Colombi, Florian Walder, Lucie Büchi, Marlies Sommer, Kexing Liu, Johan Six, Marcel G. A. van der Heijden, Raphaël Charles and Thomas Keller

Dear Editor,

We herewith submit the response to the comments by the Editor to our revised manuscript (soil-2018-35). The changes are marked in yellow in the attached manuscript file.

We hope that our adjustments are satisfactory and that the manuscript is suitable for publication in SOIL.

Sincerely,
Tino Colombi, Florian Walder, Lucie Büchi, Marlies Sommer, Kexing Liu, Johan Six, Marcel G. A. van der Heijden, Raphaël Charles and Thomas Keller

**Responses to comments by Editor**

**Editor comment:** Thank you for your detailed reply to the reviewers comments. I think that you adequately responded to their comments. But, I still have a question about the causality between the relation between aeration and soil organic matter. Can you conclude that in increase in soil aeration is the cause of an increased soil organic matter (because of higher root input)? Or can the causality be in the other direction: an increased soil organic matter content (which generates better conditions for root growth but because of other reasons than increasing soil aeration) leads to a better soil aeration? Could you give a short discussion about this?

*Response: We appreciate this valuable comment. In our opinion, the data that is included in the current study does not allow for a concluding answer to this question. This is the main reason why we emphasize that **soil structure is a crucial element** in the proposed feedback cycle (as illustrated in Fig. 5). In our opinion, beneficial effects of higher soil organic carbon content improves soil structure, which then in turn lead to better aeration, higher water holding capacity, and improved soil penetrability (i.e. soil physical conditions that foster root growth). To address and clarify this, we added a sentence to the discussion (pg 11 L28-30). Furthermore, we point out the need for experiments in which different soil physical properties can be controlled independently from soil structure. Such experiments would allow to quantify the importance of the different soil physical properties on carbon dynamics in soil (pg11 L34-pg12 L1).*

[revised manuscript text omitted]